# Induction of Resistance Against *Sclerotinia sclerotiorum* in Rapeseed by β-Ocimene Through Enhanced Production of Coniferyl Aldehyde

**DOI:** 10.3390/ijms26125678

**Published:** 2025-06-13

**Authors:** Xiang Xie, Zhonglong Yang, Wei Zhong, Hongjiang Li, Wenjing Deng, Ying Ruan, Chunlin Liu

**Affiliations:** 1Key Laboratory of Hunan Provincial on Crop Epigenetic Regulation and Development, Hunan Agricultural University, Changsha 410128, China; xie18773658581@163.com (X.X.); yangzhonglong1114@163.com (Z.Y.); 18570403205@163.com (W.Z.); lihongjiang0218@163.com (H.L.); dwj2035@163.com (W.D.); yingruan@hotmail.com (Y.R.); 2Yuelushan Laboratory, Hunan Agricultural University, Changsha 410128, China; 3College of Agronomy, Hunan Agricultural University, Changsha 410128, China

**Keywords:** β-ocimene, *Brassica napus*, induced resistance, coniferyl aldehyde, sclerotinia disease

## Abstract

Rapeseed (*Brassica napus*) is an essential oil resource, but its yield can be significantly compromised by *Sclerotinia sclerotiorum* (*S. sclerotiorum*) infection. Due to the absence of rapeseed strains that are highly or completely immune to *S. sclerotiorum*, enhancing rapeseed resistance through genetic approaches is challenging. In this study, we developed a novel method to enhance rapeseed resistance to *S. sclerotiorum* using β-ocimene. Our results demonstrated that β-ocimene treatment significantly strengthened the defense capabilities of rapeseed. β-ocimene treatment can simultaneously activate multiple defense-related signaling pathways, including jasmonic acid signaling, salicylic acid signaling, and MAPK signaling, in rapeseed, while also inducing the accumulation of secondary metabolites coniferyl aldehyde—a key secondary metabolite in the phenylpropanoid pathway critical for plant defense responses. Furthermore, applying coniferyl aldehyde to the leaves of rapeseed can remarkably enhance its resistance to sclerotinia disease. Collectively, these findings confirm that β-ocimene activates the defense system of rapeseed, elevates the content of coniferyl aldehyde, and thereby enables rapeseed to effectively combat sclerotinia disease. The metabolomics data are available via MetaboLights under the identifier MTBLS12510. In conclusion, this study not only uncovers the mechanism by which β-ocimene induces rapeseed resistance to sclerotinia disease but also presents a novel approach for its prevention and control.

## 1. Introduction

Rapeseed (*Brassica napus*) is a vital oilseed crop with significant dietary and industrial applications [1]. However, its yield is subject to various limitations, with microbial pathogens being one of the most devastating factors [2]. Among these pathogens, *Sclerotinia sclerotiorum* (*S. sclerotiorum*), which leads to the presence of water-soaked rot and white mycelium in host rapeseed by infecting the stems, leaves, and flowers, ultimately resulting in plant death, is the pathogenic fungus responsible for Sclerotinia Stem Rot (SSR) of rapeseed [3]. Due to the long survival term of the sclerotia of *S. sclerotiorum* in the soil, *S. sclerotiorum* infestation in rapeseed can be high in occurrence under favorable environmental conditions [4]. Nevertheless, the lack of the rapeseed germplasm being completely immune to SSR makes it challenging to clone the SSR resistant genes from rapeseed [5]. Given the limited progress in developing highly or completely immune rapeseed strains through genetic breeding or engineering, the need for novel approaches to combat SSR has become increasingly urgent [5].

Volatile compounds (VOCs) emitted by plants are a group of bioactive substances functionally essential for plant interactions [6]. Recently, accumulating evidence has suggested the role of VOCs in inducing plant immune response [7,8]. Ocimene, a VOC predominantly synthesized by terpene synthases (TPS) [9], potentially facilitates plant defense response through both indirect and direct mechanisms [10,11,12,13]. Indirectly, β-Ocimene might act as an herbivore-induced volatile that altruistically primes neighboring plants through enabling the tea plants to efficiently repel female herbivores [11]. And in damaged asparagus, ocimene protects the plant by attracting the parasitoids to the pests [12]. Ocimene has been shown to directly activate defense response pathways in plants at both biochemical and molecular levels. For instances, ocimene protects *Arabidopsis thaliana* from the feeding of *Spodoptera litura* through modulating the histone acetyl status of the genes encoding ethylene-responsive factors ERF8 and ERF104. And the defense response induced by ocimene can last for 5 to 10 days [10]. Furthermore, the ocimene-induced transcriptome of *Arabidopsis thaliana* closely resembles the transcriptional patterns observed following methyl jasmonate (MeJA) treatment, mechanical wounding, and insect feeding [13]. It has been shown that ocimene treatment inhibited Botrytis cinerea mycelium growth on Arabidopsis leaves [14]. When Columbia wild-type Arabidopsis was inoculated with Pseudomonas syringae following β-ocimene treatment, leaf necrosis was less severe compared to that of the untreated group [15]. In addition, β-ocimene can alleviate the antagonism between salicylic acid (SA) and jasmonic acid (JA), enabling the co-expression of resistance-related genes *PR1* and *PDF1.2*, while also altering secondary metabolites. Therefore, Ocimene is a VOC presenting versatile roles in activating the defense response in multiple types of plants [16,17,18]. However, whether ocimene modulates the ability of rapeseed to resist SSR and the underlying mechanisms remains to be identified.

Through the integration of transcriptomic and metabolomic profiling, our study identified key transcripts and metabolites altered by β-ocimene, which enhance rapeseed’s defense response against *S. sclerotiorum* infection. We showed that ocimene treatment evidently protected the susceptible rapeseed variety 98C40 from *S. sclerotiorum* infestation. Our transcriptomic and metabolic profiling results indicated the activation of multiple signaling pathways, including the phenylpropanoid pathway, jasmonic acid (JA) pathway, and salicylic acid (SA) pathways, in β-ocimene-treated groups upon *S. sclerotiorum* inoculation, suggesting the potential mechanisms by which β-ocimene directly triggers the immune response in rapeseed. More importantly, we noticed that the production of coniferyl aldehyde, which is a key component of the phenylpropanoid pathway in plant defense response [19], was drastically elevated following the β-ocimene treatment. Furthermore, our functional experiments provided additional validation of the hypothesis. Based on our transcriptomic and metabolic data, we demonstrated that β-ocimene treatment significantly enhanced rapeseed resistance to *S*. *sclerotiorum* infection by altering its transcriptome and metabolism. Furthermore, our findings suggested that coniferyl aldehyde accumulation, induced by β-ocimene, played a crucial role in strengthening rapeseed’s defense against Sclerotinia disease. Our study elucidated the physiological mechanisms through which β-ocimene enhanced rapeseed resistance and provided valuable insights into the development of β-ocimene-based strategies for the prevention and control of Sclerotinia disease.

## 2. Results

### 2.1. β-Ocimene Enhanced the Resistance of Rapeseed to S. sclerotiorum

Based on our pilot studies, the optimized β-ocimene concentration to treat rapeseed was 15 μM. At this concentration, β-ocimene simultaneously induced the gene expression of *PR1* and *PDF1.2*, resulting in the strongest resistance of rapeseed (Appendix A). To investigate the function of β-ocimene in the resistance of rapeseed to *S. sclerotiorum*, we treated the seedlings of susceptible *Brassica napus* variety 98C40 with 15 μm β-ocimene for 12 h, then inoculating *S. sclerotiorum* agar blocks on the leaves. We observed significant differences in the infection of rapeseed by *S. sclerotiorum* between the control group and treated group (Figure 1A). Statistically, the disease area on the treated plants was reduced by 90.96% to 82.19%, compared with the control group (Figure 1B). Consistently, trypan blue staining (Figure 1C) indicated that mycelial invasion and necrosis in the treated group were less severe than those in the control group, suggesting β-ocimene treatment enhanced the defense response and protected rapeseed from cell damage caused by *S. sclerotiorum* infection. Quantification of *S. sclerotiorum* in the inoculated leaf area revealed that the pathogen’s DNA content in the control group was more than ten times higher than that in the treated group (Figure 1D).

### 2.2. Transcriptomic and Metabolic Profiling of β-Ocimene-Treated Rapeseed

To uncover the molecular mechanisms underlying rapeseed resistance to *S. sclerotiorum* following β-ocimene treatment, transcriptomic profiling was conducted on the fifth leaves of the treated and control plants. In total, six samples were subjected to sequencing, including three biological replicates for the control and treated groups. Gene expression levels were analyzed by calculating transcripts per million (TPM) (Figure 2A). The first two axes of the Principal Component Analysis (PCA) showed that 95.04% of the variation among the samples was due to differences in the TPM (Figure 2B), suggesting our data quality was sufficient for subsequent analysis. DEG analysis suggested that there were 767 up-regulated and 203 down-regulated genes in the treated group (Figure 2C). KEGG enrichment analysis identified that the DEGs between the treated and control groups were dominantly enriched in the pathways such as ribosome biogenesis in eukaryotes, anthocyanin biosynthesis, galactose metabolism, plant-pathogen interaction, sulfur metabolism, plant hormone signal transduction, and RNA degradation (Figure 2D).

In addition to transcriptomic profiling, we also conducted metabolic profiling of β-ocimene-treated rapeseed. Specifically, 25,922 compounds, including 12,640 primary metabolites and 652 secondary metabolites were detected by metabolic profiling. The secondary metabolites were affiliated to 15 classes of compounds, including lipids and lipid-like molecules, organic acids and their derivatives, organic heterocyclic compounds, benzene compounds, phenylpropanoids and polyketides, organic oxides, nucleosides and nucleotide analogs, alkaloids and their derivatives, organic oxides, lignin and neolignans and related compounds, sulfonamides, organic nitrogen compounds, organic halogen compounds, and organic sulfur compounds. The PCA plots of the negative ion mode and positive ion mode showed dispersion between the control and treated groups but no dispersion within biological replicates (Figure 3B). The PCC for the negative ion mode was higher than 0.68, and the PCC for the positive ion mode was higher than 0.9 (Figure 3B). These results suggested that our data was high quality. Further analysis showed that 29 metabolites were up-regulated and 24 metabolites were down-regulated in the treated group (Figure 3A). KEGG enrichment analysis of the altered metabolites indicated changes in several pathways within the treated group (Figure 3C,D). By highlighting the top 30 metabolites in the heatmap, we showed that upon β-ocimene treatment, metabolites such as ibuprofen, oxidized glutathione, phosphatidylcholine 50:9, phosphatidylcholine 56:11, oxalic acid, 3,4-dimethylphenol, phosphatidylglycerol 33:3, fumaric acid, stearoyl-lyso-phosphatidylglycerol, phosphatidylglycerol 35:4, tri-(4-tert-butyl-3-hydroxy-2,6-dimethylbenzyl), isocyanuric acid ester, 13-oxo-9,11-tridecenoic acid, riboflavin-5′-monophosphate, methotrexate, S-adenosyl-L-homocysteine, and flucarbazone were down-regulated. Metabolites such as dehydro-1,8-cineole, valganciclovir, cytidine, 2,3-dihydro-3,3,5,6-tetramethyl-1(1H)-indanone, L-methionine, L-tyrosine, phosphatidic acid 33:3, 6-methylquinoline, tryptophan, (±)-tryptophan 2-methylindole, indole, 2-aminonaphthalene, and quinoline-4,8-diol were up-regulated (Figure 3E).

More importantly, integrated analysis of transcriptomic and metabolic profiling revealed an enhanced production of coniferyl aldehyde in β-ocimene-treated rapeseed. A Venn analysis of the KEGG pathways enriched with differential metabolites from each omics dataset suggested there were 20 common KEGG pathways between transcriptomic and metabolic profiling results (Figure 4A). And the common pathways related to plant resistance to *S. sclerotiorum* included the phenylpropanoid biosynthesis pathway, glucosinolate metabolism pathway, secondary metabolite biosynthesis pathway, and oxalate biosynthesis pathway (Figure 4B). More importantly, when we delved into the differential metabolites within these common KEGG pathways (Figure 4C–E), we noticed that coniferyl aldehyde, a key component of the phenylpropanoid pathway in plant defense response, was drastically increased in the treated group. Therefore, we hypothesized that β-ocimene-induced up-regulation of coniferyl aldehyde was crucial for rapeseed in resisting SSR.

Based on Figure 4B, the sulfur metabolism and glucosylation biosynthesis pathways are more enriched than the phenylpropanoid biosynthesis of pinocembrin. The latter compound is more focused on gluconates, whose degradation products (i.e., isothiocyanates) are known antimicrobial agents.

In Figure 4E, pinocembrin is up-regulated by a 1.65-fold change (FC), while 1-isothiocyanato-4-(methylsulfinyl)-butane is up-regulated by a 1.44-fold change (FC).

However, since our goal is to identify a novel metabolite significantly altered by β-ocimene induction—an area that remains relatively unexplored in resistance research—this study will prioritize coniferyl aldehyde as the primary target moving forward.

### 2.3. Validation of DEGs by RT-Quantitative Real-Time PCR (qPCR)

To validate the DEGs between the control and β-ocimene-treated rapeseed identified by transcriptomic profiling, we examined the mRNA levels of specific genes involved in the SA and JA signaling pathways by RT-qPCR. We treated 98C40 with β-ocimene, SA, and JA, respectively. Then the samples were subjected to RT-qPCR to quantify the mRNA levels of *PR1*, *PDF1.2*, and *WRKY70*, all of which are the genes encoding the components of the SA and JA pathways. We showed that the expression levels of these three genes were significantly increased in β-ocimene-treated rapeseed, compared to those in the control group after ocimene treatment (Figure 5A).

Moreover, we randomly selected three genes from the DEGs generated from transcriptomic profiling for RT-qPCR to further validate our transcriptomic data. These three genes with unknown functions were LOC106364097, CYP45094C1 (LOC106391362), and ERF025 (LOC111209820). Our RT-qPCR suggested that the relative expression levels of these genes in treated rapeseed were significantly higher than those of the control group (Figure 5B), which was in line with our transcriptomic analysis.

### 2.4. Coniferyl Aldehyde Treatment Enhanced the Resistance of Rapeseed to S. sclerotiorum

While individual transcriptomic or metabolic profiling did not show an effect of ocimene on coniferyl aldehyde metabolism, coniferyl aldehyde, the precursor of lignin, has been identified as a differential metabolite through integrated omics analysis. To prove that β-ocimene-induced up-regulation of coniferyl aldehyde was able to aid rapeseed in combating *S. sclerotiorum* infection and *Spodoptera litura* feeding, we treated the leaves of rapeseed with coniferyl aldehyde dissolved in ethyl acetate. Our results showed that the treatment of coniferyl aldehyde significantly repressed the infection of rapeseed by *S. sclerotiorum* (Figure 6A,B).

## 3. Discussion

It is well established that plants respond to pathogen infections by synthesizing a diverse array of secondary metabolites with antimicrobial properties, effectively inhibiting the growth and spread of *S. sclerotiorum* [20]. Plants can enhance their physical barriers through thickening the cell wall and depositing lignin to prevent the invasion of Sclerotinia. Specifically, the accumulation of secondary metabolites such as flavonoids and lignin precursors can strengthen the plant cell wall, thereby hindering pathogen expansion [20]. In line with these previous findings, in this study we observed that coniferyl aldehyde, one of the lignin precursors, accumulated in the β-ocimene-treated leaves that had an intensive ability to resist bacteria invasion (Figure 4C–E). More importantly, our functional assay also confirmed that the accumulation of coniferyl aldehyde enhanced the resistance of rapeseed to bacteria and pests (Figure 6), which was consistent with the phenotype we observed from β-ocimene-treated rapeseed (Figure 1), further indicating that coniferyl aldehyde accumulation potentially explained the enhanced defense response triggered by β-ocimene. Nevertheless, further direct evidence is required to definitely establish that β-ocimene enhances rapeseed resistance to *S. sclerotiorum* through the promotion of coniferyl aldehyde production. Future studies should investigate whether inhibiting coniferyl aldehyde production would compromise the SSR resistance capability conferred by β-ocimene in rapeseed.

In addition to coniferyl aldehyde, we also reported a variety of signaling pathways proven to be involved in plant defense response were activated in β-ocimene-treated rapeseed. For instance, the glucosinolate metabolic pathway plays an important role in the disease resistance response of cruciferous plants, with its degradation products like isothiocyanates presenting potent antimicrobial activity [21]. Therefore, isothiocyanates could be a potential metabolic candidate in the resistance of rapeseed to bacteria. Here, we showed the up-regulation of methionine and tryptophan in β-ocimene-treated rapeseed (Figure 3E and Figure 4C), both of which are crucial intermediates in glucosinolate synthesis. In addition, the jasmonic acid (JA) signaling pathway promotes the synthesis of antimicrobial substances, such as proteinase inhibitors and secondary metabolites, by regulating the defense-responsive genes [22]. And salicylic acid (SA) is the key functional molecule in plant systemic acquired resistance (SAR), important for the plant defense against biotic stress, such as fungi and bacteria [23]. Mechanistically, SA signaling can activate defense-responsive genes, like *PR* and *NPR1*, thereby elevating the plant’s resistance to pathogens [23]. In this study, we also showed the co-activation of JA and SA signaling in the plants treated with β-ocimene (Figure 2D), which was in line with the previous findings [22,23,24]. Moreover, we also demonstrated the activation of MAPK signaling in the treated plants (Figure 2D), which has been suggested to help plants in resisting *S. sclerotiorum* [25,26]. Collectively, our omics data indicated that β-ocimene activated multiple defense response pathways, enhancing the resistance of rapeseed to SSR.

Overall, our study suggests that β-ocimene plays a crucial role in enhancing the defense response of rapeseed. This enhancement occurs through the accumulation of secondary metabolites with antimicrobial properties and the activation of defense-responsive signaling pathways. By accumulating these secondary metabolites, rapeseed plants become better equipped to combat pathogen attacks and reinforce their overall defense mechanisms. Our findings provide novel insights into the potential applications of volatile compounds like β-ocimene in sustainable crop protection strategies. Utilizing such compounds can offer a natural and environmentally friendly approach to enhancing plant resistance against various pathogens. This could reduce the reliance on synthetic chemicals and promote healthier and more resilient crops. Moreover, our research opens up new avenues for exploring the use of other volatile compounds in agriculture. By elucidating the mechanisms through which these compounds enhance plant defense responses, we can develop innovative and sustainable strategies to protect crops, ensuring both food security and environmental sustainability. These insights underscore the importance of further research to explore the full potential of volatile compounds in crop protection. Future studies could explore additional volatile compounds with similar or enhanced effects on plant defense while optimizing their application in agricultural practices.

## 4. Materials and Methods

### 4.1. Plant Materials and Growth Conditions

The susceptible *Brassica napus* variety 98C40 from the Key Laboratory of Epigenetics and Development of Hunan Province, Hunan Agricultural University was used for this study. After germinating for two days in the dark at 23 °C, the seedlings were transplanted into nutrient soil and grown in a growth chamber with a 16 h light/8 h dark photoperiod (16 h light, 22 °C; 8 h dark, 20 °C). The inoculated leaves were selected from the third or fifth leaves of four-week-old plants.

Based on our thoroughly phenotypic experiments, we collected the samples for transcriptomic and metabolic profiling at the time points that β-ocimene treatment can lead to significant phenotypic differences in the resistance of rapeseed post-*S. sclerotiorum* inoculation. After treatment, the same batch of samples was divided into two groups: one was cryopreserved in liquid nitrogen, and the other one was subjected to *S. sclerotiorum* inoculation and gene expression analysis for *PR1* and *PDF1.2*. Samples with evident phenotypes and co-expression of marker genes were qualified for transcriptomic and metabolic profiling.

### 4.2. β-Ocimene Treatment

The drying jars were cleaned and dried and then disinfected with 75% ethanol. Materials from the control group and treated group were placed separately in the jars. β-ocimene was added to the glass sheet in the drying jars to achieve a final concentration of 15 μmol/L. The jars were sealed with tape and transferred to an artificial climatic chamber (16 h light/8 h dark photoperiod, light intensity 100 μmol·m^2^·s^−1^) for 12 h. After treatment, the materials were transferred to clean, disinfected drying jars to remove the remaining β-ocimene and were then prepared for inoculation. Salicylic acid (MW: 138.12 g/mol) was prepared by mixing 0.0138 g powder with 1 mL ethanol, 49 mL water, and 7.5 μL L77 to achieve a 2 mM solution (total volume: 50 mL). The working concentration was 2 mM. Methyl jasmonate (MW: 224.3 g/mol, purity: 95%) was prepared by mixing 100 μL methyl jasmonate with 218.4 mL pure water and 32.76 μL L77, and stirring to mix thoroughly. The working concentration was 0.2 mM. The treatments were evenly sprayed onto the leaves and backs of the plants, with a duration of 3 h.

### 4.3. Treatment of Coniferyl Aldehyde

An amount of 100 mg coniferyl aldehyde was dissolved in 2 mL ethyl acetate solution, and Tween-2 with 0.5% volume concentration was added to 25 mL emulsion and diluted with water to a final concentration of 4 mg/mL. The leaves of the rapeseed were uniformly oscillated before spraying. The control group and treatment group were set.

### 4.4. Evaluation of Sclerotinia Resistance

Sclerotinia isolates SS-1 were cultured on potato dextrose agar (Solarbio, Beijing, China) at 21 °C in the dark. A 9 or 7 mm mycelial agar plug from the vigorously growing edge of a 5-day-old culture was used to inoculate plant leaves. After inoculation, the plants were placed in sealed drying jars and transferred to an artificial climatic chamber (16 h light/8 h dark photoperiod, light intensity 100 μmol·m^2^·s^−1^). Leaves were photographed, and the lesion area was quantified at the symptoms peak by ImageJ 1.53H. ImageJ 1.53H was used to select lesions and calculate their sizes, with the average value obtained from multiple measurements. Leaf DNA was extracted from the area (3 cm × 4 cm) near the inoculation site, and quantitative PCR was used to detect the expression of *SS-TY* as a measurement of fungal colonization. The following primers were used. TY-F: ATATAACGCTACTCTCTCTGTTC; TY-R: AGCCAACTTTCGGAGATTTG.

### 4.5. Metabolite Profiling

The fifth leaves of normal 98C40 were used for metabolomics analysis. The treated group was labeled as Oci, and the control group was labeled as Mock. Six plants were randomly selected from each group, rapidly frozen in liquid nitrogen, and stored at −80 °C for later use. The samples were thawed on ice, and metabolites were extracted using 50% methanol buffer. Briefly, 100 mg of sample was extracted with 1 mL of pre-cooled 50% methanol, vortexed for 1 minute, and incubated at room temperature for 10 minutes. The extract was stored overnight at −20 °C. After centrifugation at 4000× *g* for 20 min, the supernatant was transferred to a new 96-well plate. The samples were stored at −80 °C prior to the use of a Liquid Chromatograph/Mass Spectrometer (LC-MS). The metabolites eluted from the column were detected by a high-resolution tandem mass spectrometer TripleTOF 6600 (SCIEX, Framingham, MA, USA). All samples were collected by an LC-MS according to the manufacturer’s instructions. All chromatographic separations were performed using the UltiMate 3000 UPLC system (Thermo Fisher Scientific, Bremen, Germany). Reverse-phase separation was carried out using an ACQUITY UPLC T3 column (100 mm × 2.1 mm, 1.8 μm, Waters, Milford, CT, USA). The column oven was maintained at 40 °C, with solvent A (5 mM ammonium acetate and 5 mM acetic acid) and solvent B (acetonitrile). The low flow rate was set to 0.3 mL/min with solvent A as the mobile phase. The gradient elution conditions were set as follows: 0–0.8 min, 2% B; 0.8–2.8 min, 2–70% B; 2.8–5.6 min, 70–90% B; 5.6–6.4 min, 90–100% B; 6.4–8.0 min, 100% B; 8.0–8.1 min, 100–2% B; 8.1–10 min, 2% B.

An amount of 10 μL of each extract was pooled to prepare mixed QC samples. The extracted samples were randomly ordered for analysis, with QC samples inserted before, during, and after sample analysis to evaluate experimental repeatability. The samples were subjected to mass spectrometry analysis in both positive and negative ion modes. Differential metabolites were identified with the following criteria: FC ≥ 1.2 or FC ≤ 1/1.2, *p*-value < 0.05, VIP ≥ 1. Differentially expressed metabolites were annotated, enriched, and mapped to KEGG (Kyoto Encyclopedia of Genes and Genomes) pathways. The metabolomics data have been deposited to MetaboLights repository with the study identifier MTBLS12510.

### 4.6. RNA Extraction, Library Construction, and Sequencing

The fifth leaves from normal 98C40 were used for transcriptomic profiling. The treated group was labeled as Oci, and the control group was labeled as Mock. Three plants were randomly selected from each group, rapidly frozen in liquid nitrogen, and stored at −80 °C for later use. RNA was extracted from the samples using TRIzol® (Thermo Fisher, 15596018) according to the manufacturer’s instructions. The quantity and purity of the total RNA were assessed using a NanoDrop ND-1000 (NanoDrop, Wilmington, DE, USA), and RNA integrity was evaluated with a Bioanalyzer 2100 (Agilent, Santa Clara, CA, USA). The samples with a concentration > 50 ng/μL, RIN value > 7.0, and total RNA > 1 μg were used for downstream experiments. Poly(A)+ mRNA was captured by oligo(dT) magnetic beads (Dynabeads Oligo (dT), cat.25-61005, Thermo Fisher, Waltham, MA, USA) through two rounds of purification. The captured mRNA was fragmented at high temperature using a magnesium ion fragmentation kit (NEBNextR Magnesium RNA Fragmentation Module, cat.E6150S, Ipswich, MA, USA), generating a library with fragment sizes of 300 bp ± 50 bp (strand-specific library). Finally, the library was sequenced using the Illumina NovaSeqTM 6000 (LC Bio Technology Co., Ltd., Hangzhou, China) with a paired-end sequencing mode of PE150 according to standard procedures. Total RNA extraction, quality assessment, cDNA library construction, and sequencing were performed by LC Bio Technology Co., Ltd. Differentially Expressed Genes (DEGs) between the control and treated groups analysis were analyzed by DESeq2, with the criteria for identifying differentially expressed genes being |log2(fc)| ≥ 1 & q < 0.05.

FastQC 0.10.1, Hisat2 2.2.1, StringTie 2.1.6, DeSeq2/edgeR 1.22.2/3.22.5, R 3.6, rMATS 4.1.1 were used for quality control, genomic BLAST, transcriptomic assembly and quantification, differential gene analysis, plotting, and splicing analysis, respectively.

### 4.7. Differential Gene Screening and RT-qPCR Analyses

DESeq2 was used for differential expression analysis, with the criteria of |log2(fc)| ≥ 1 and q < 0.05 to identify differentially expressed genes. For RT-qPCR analysis, relative gene expression levels were calculated by 2∆∆Ct. The primers used for RT-qPCR are listed in Table 1.

### 4.8. Statistical Analysis and Other Materials

GraphPad Prism 10 and Excel 2010 were used for graphing, data statistical analysis, and one-way ANOVA (*p* = 0.05), with LSD method for comparisons. The following reagents and materials were used in this study: 82% β-Ocimene (Sigma-Aldrich, Taufkirchen, Germany), 75% ethanol, 20% sodium hypochlorite solution, pure water, Potato Dextrose Agar (PDA) medium (Solarbio), Trypan Blue staining solution (Solarbio), (E) 3,3′-Diaminobenzidine (DAB) (Coolaber), Coniferaldehyde powder (leaf-derived, Shanghai). The following equipments were used in this study: HST-12 Artificial Climate Chamber (Shanghai Shiteng Instruments, Shanghai, China), GZ-300-S Biochemical Incubator (Shaoguan Guangzhi Technology Equipment Co., Ltd., Guangdong, China), Digital SLR high-definition camera (Nikon Corporation, Tokyo, Japan). Omics sequencing was performed by Lianchuan-Bio (Hangzhou, China).

## Figures and Tables

**Figure 1 ijms-26-05678-f001:**
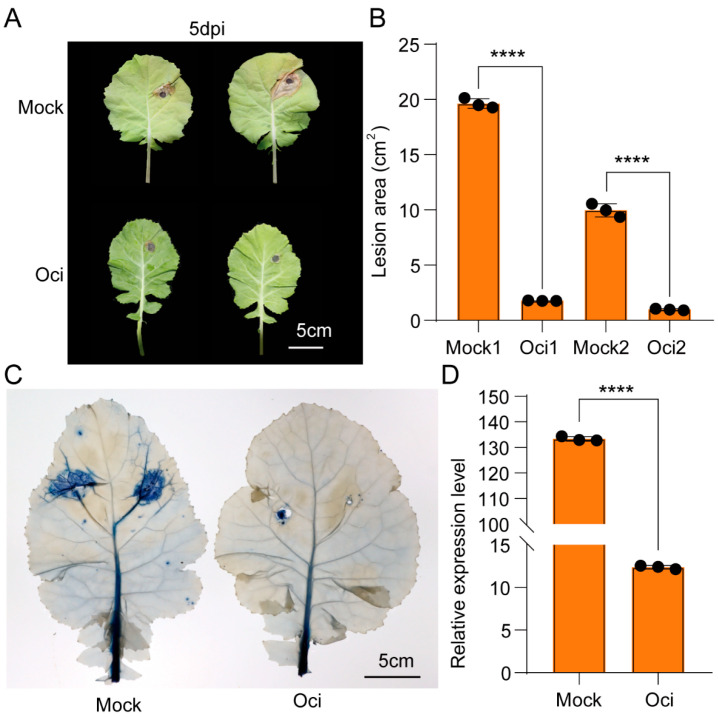
β-ocimene treatment enhanced the resistance of 98C40 to *S. sclerotiorum* infection and *Spodoptera litura* feeding. (**A**) Representative photos of the leaves infected by *S. sclerotiorum*. (**B**) Quantification of the lesion area on the leaves. (**C**) Quantification of cell necrosis by trypan blue staining. (**D**) Quantification of pathogen DNA in the area adjacent to 9 mm mycelium agar block leaves at 5 dpi by quantitative real-time PCR (qPCR). Mock: control group; Oci treated group; scale bar: 5 cm. Student’s *t* test was used to determine the significant difference; the symbols **** indicate significant differences at the *p* < 0.0001 level. Error bars represent SD.

**Figure 2 ijms-26-05678-f002:**
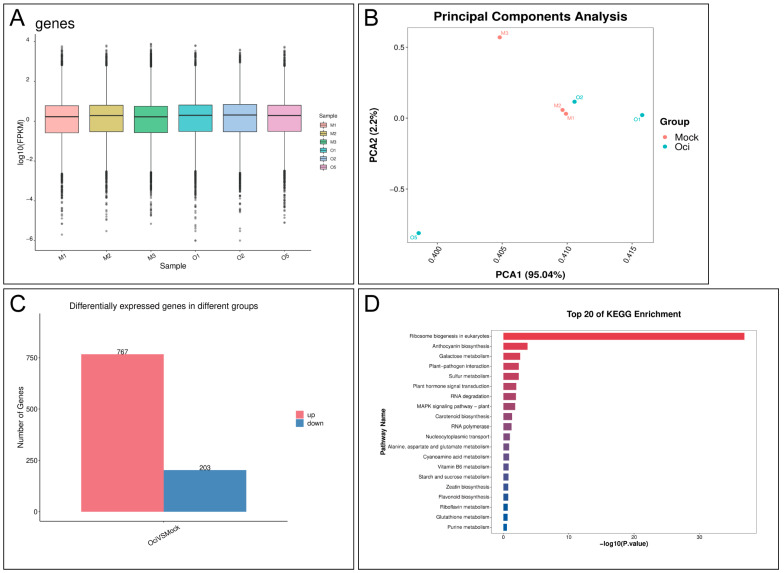
Transcriptomic analysis of β-ocimene-treated and untreated 98C40. (**A**) Distribution of average TPM. (**B**) PCA. (**C**) Summary of DEGs between β-ocimene-treated and untreated groups. (**D**) Top 20 pathways from KEGG enrichment. Mock: control group; Oci: treated group.

**Figure 3 ijms-26-05678-f003:**
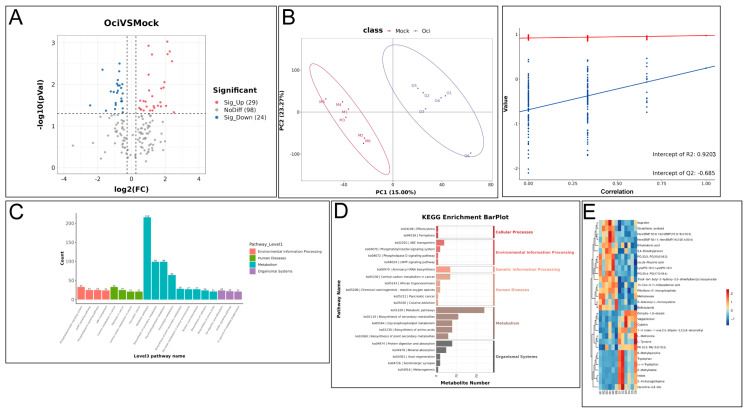
Metabolic analysis of β-ocimene-treated and untreated 98C40. (**A**) Volcano plot showing overall distribution of differential metabolites between β-ocimene-treated and untreated groups. (**B**) PCA. (**C**) KEGG enrichment barplot. (**D**) Top 20 pathways from KEGG enrichment. (**E**) Heatmap showing top 30 differential metabolites between β-ocimene-treated and untreated groups. Mock: control group; OCI: treated group.

**Figure 4 ijms-26-05678-f004:**
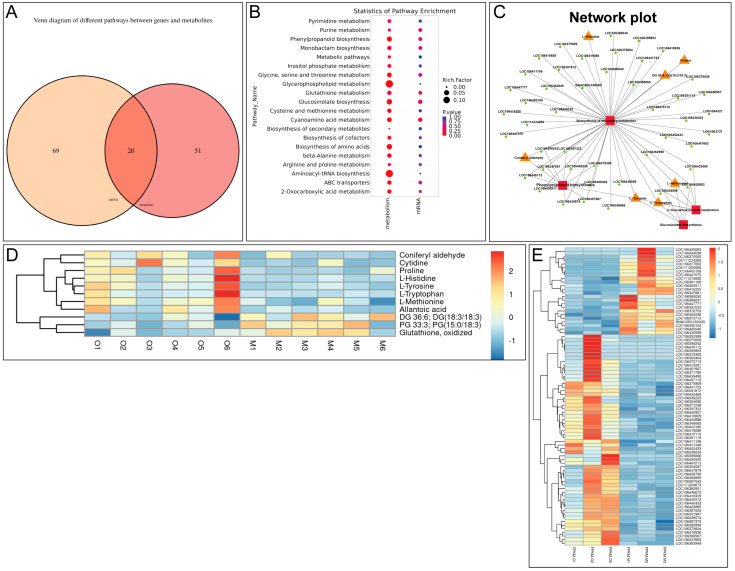
Joint analysis of transcriptomic and metabolic profiling. (**A**) Venn diagram showing KEGG pathways enriched with differential metabolites from each omics dataset. (**B**) Multi-group bubble diagrams showing pathways enriched in transcriptomic and metabolic profiling. (**C**) Metabolite-related sub-network diagram. (**D**) Heatmap showing DEGs between β-ocimene-treated and untreated 98C40 identified by transcriptomic profiling. (**E**) Heatmap showing differential metabolites between β-ocimene-treated and untreated 98C40 identified by metabolic profiling. Mock: control group; OCI: treated group.

**Figure 5 ijms-26-05678-f005:**
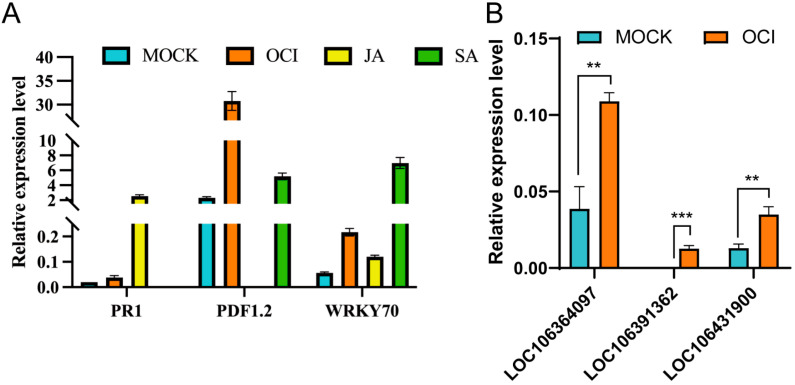
(**A**) RT-qPCR analysis of *PR1*, *PDF1.2*, and *WRKY70* in 98C40 treated with β-ocimene, SA, and JA, respectively. (**B**) RT-qPCR analysis of three genes with unknown functions in 98C40 treated with β-ocimene; two-factor analysis of variance was used. The symbols **, *** indicate significant differences at the *p* < 0.01, *p* < 0.001 level. Error bars represent SD.

**Figure 6 ijms-26-05678-f006:**
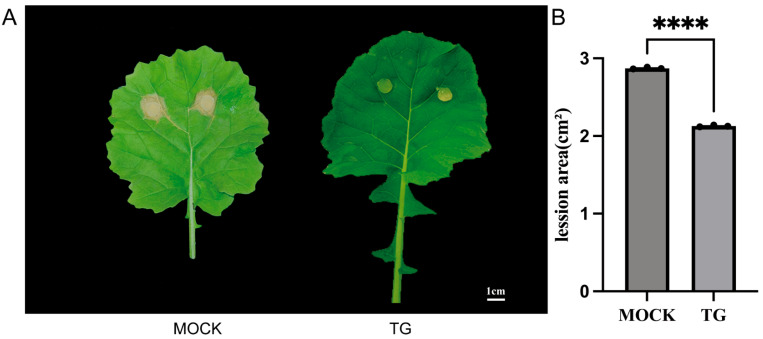
Coniferyl aldehyde treatment enhanced the resistance of 98C40 to *S. sclerotiorum* infection and *Spodoptera litura* feeding. (**A**) Representative photos of the coniferyl aldehyde-treated and untreated leaves that were infected by *S. sclerotiorum*. The final concentrations of coniferyl aldehyde were 2 mg/mL, 4 mg/mL, and 8 mg/mL. (**B**) Lession area of the TG-treated or TG-untreated 98C40 infected by *S. sclerotiorum.* Student’s *t* test was used to determine the significant difference; **** indicate significant differences at the *p* < 0.0001 level. Error bars represent SD.

**Table 1 ijms-26-05678-t001:** List of RT-qPCR primers.

Gene	Annotation	Forward Primer	Reverse Primer
*BnaPR1*	PR1	GCTCTTGTTCATCCCTCGAAAGC	GTCGGCGTAGTTTTGAGCGTAG
*BnaPDF1.2*	PDF1.2	CATCACCCTTCTCTTCGCTGC	ATGTCCCACTTGACCTCTCGC
*BnaWRKY70*	WRYK70	ACATACATAGGAAACCACACG	ACTTGGACTATCTTCAGAATGC
*LOC106364097*	IQ domain-containing protein IQM1	TTGACCCACGCCATCGTTAT	AATGGCTGTGAGCTCTTGCT
*LOC106391362*	cytochrome P450 94C1-like	ATGCATTATTTGCATGCGGC	CGTACCATCAGCCAAGACG
*LOC106431900*	ERF113	TCAAGGCCCATCAACCACAA	ACTACCTCCTCTTGGGGCAT
*ACTIN7*	ACTIN7	GACAATGGAACTGGAATGGTGAAGG	GTTGCTTACAATACCATGCTCAATCGG

## Data Availability

The original contributions presented in this study are included in the article and Appendix A. Further inquiries can be directed to the corresponding author.

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
