# Peer review of "Induction of Resistance Against Sclerotinia sclerotiorum in Rapeseed by β-Ocimene Through Enhanced Production of Coniferyl Aldehyde"

_ijms, 2025, doi:10.3390/ijms26125678_

Round 1
Reviewer 1 Report
Comments and Suggestions for Authors
- In the part of Introduction, most cited research focus on the role of ocimene in pest resistance in plants, and its role in disease resistance should be enhanced.
- In the last part of Introduction, it should highlight the objective and significance of this study, instead of the results.
- Why 15 uM of ocimene was selected, is there any other concentrations have been applied?
- How to determine the time of trancriptome and metabolism analysis.
- The effects of ocimene on the metabolism of coniferyl aldehyde were not clearly demonstrated based on the data of transcriptome and metabolism analysis.
- Some of the figures are two small with low resolution, and there is no figure D in Figure 1.
- In the measurement of lession area, the data of diffrent replicates were too close. How was the lession area counted?
- The language should be improved.
Reviewer 2 Report
Comments and Suggestions for Authors
The manuscript focus on the treatment of Rapeseed (Brassica napus) leaves with β-ocimene to prevent and reduce the infection of Sclerotinia sclerotiorum.
While the entire introduction and presentation of the results fit with a logic flow of the presentation of the various findings about the differential metabolomics and transcriptomics studies, the figures are all affected by the small resolution that impair the reading of the candidate metabolites/genes.
It is advisable to upload high resolution images that allow the reading of small details.
In Figure 1, the panel B has missing legend (e.g. oci1 is truncated to oci and the "M" of MOCK is also halp.
In Figure 4. the panel D and E are swapped.
Check the neglish usage: "ditermined the signifficant" this error is repeated over the manuscript...
As conclusion, not only the coniferyl aldehyde as final effector of β-ocimene treatment but also the glucosinolate metabolic pathway that is also upregulated could lead to the accumulation of glucosinolates with antifungal fuction as described. NO info on the glucosinolate accumulating metabolite is provided because no supplementary files is given on the sucessful identified metabolites and their differential amount on the MOCK control as compared the β-ocimene treatment.
The methods part is lacking of data of Metabolite profiling: such as which LC-MS system and condition that they have used for the runs, supplementary files that demonstrate the obtaining of differential metabolites. KEGG related pathway metabolite, differential analysis and heatmap analysis related to Figure 3 and 4.
The same for the transcriptomics analysis, no info on the raw data accessions (e.g. SRA) or supplementary files demonstrating the DEG analysis by DESeq2 alghoritm or the KEGG enrichment analysis (Figure 2). Besides, no supplementary file is given for the Heatmap showing the DEGs (Figure 4).
Comments on the Quality of English LanguageIn lane 15: the "strain" adjective is used for fungi and not for plants. For plants the "variety" or "cultival" term is used instead.
Some error : "ditermined the signifficant" is repeated.
Also errors might be present....
Round 2
Reviewer 1 Report
Comments and Suggestions for Authors
The manuscript has been revised as recommended.
Reviewer 2 Report
Comments and Suggestions for Authors
The authors have made some changes in the manuscript text that improve the readibility and the comprehension of the procedured. Supplementary materials were necessary to undersatnd the details of the figures. However, no high-res figures were supplied in this revision. Besides, still metabolomics and transcriptmocis data are missing as raw data downloadable link. No Mass spectometer instrument is mentioned, only the LC details are presents. The metabolomics data are not present in the MetaboLights database either by searching tusing the identifier MTBLS12510 or entering Brassica napus as organism (https://www.ebi.ac.uk/metabolights/search?organism.organismName=Brassica%20napus).
The same for the SRA raw transcriptomics data: no link is given in the manuscript.
Besides no RNA/Brassica napus reference genome aligner is mentioned in material and methods, since the DESeq2 methods apply to the bam files generated through it.
From figure 4 supplementary data (differential expresed genes) the Sufur Metabolism and the Glucosinolate biosynthesis pathway are also enriched more than the Phenylpropanoid biosynthesis were the coniferyl aldehyde. Still this last compound has got to you more attention of glucosinolates which degradation products (i.e. isothiocyanates) are known antimicrobial agents.
For instance, looking into the supplementary file for figure 4 (metabolomic differential compounds): 1-Isothiocyanato-4-(methylsulfinyl)-butane is upregulated 1.44 fold change (FC) as compared to Coniferyl aldehyde that is also upregulated with 1.65 FC.
Nevertheless, more importance is given to the Coniferyl aldehyde/β-ocimene than the isothiocyanates. Please write few lines in the discussion on the possible co-involvement of the isothiocyanates too for this pathogen, perhaps involving it as hypothesis of next screening.
